# Molecular Pathogenesis and Regulation of the *miR-29-3p*-Family: Involvement of *ITGA6* and *ITGB1* in Intra-Hepatic Cholangiocarcinoma

**DOI:** 10.3390/cancers13112804

**Published:** 2021-06-04

**Authors:** Yuto Hozaka, Naohiko Seki, Takako Tanaka, Shunichi Asai, Shogo Moriya, Tetsuya Idichi, Masumi Wada, Kiyonori Tanoue, Yota Kawasaki, Yuko Mataki, Hiroshi Kurahara, Takao Ohtsuka

**Affiliations:** 1Department of Digestive Surgery, Breast and Thyroid Surgery, Graduate School of Medical and Dental Sciences, Kagoshima University, Kagoshima 890-8520, Japan; k6958371@kadai.jp (Y.H.); k5495007@kadai.jp (T.T.); k3352693@kadai.jp (T.I.); k8911571@kadai.jp (M.W.); wilson@m.kufm.kagoshima-u.ac.jp (K.T.); k5968102@kadai.jp (Y.K.); mataki@m.kufm.kagoshima-u.ac.jp (Y.M.); h-krhr@m3.kufm.kagoshima-u.ac.jp (H.K.); takao-o@kufm.kagoshima-u.ac.jp (T.O.); 2Department of Functional Genomics, Graduate School of Medicine, Chiba University, Chiba 260-8670, Japan; cada5015@chiba-u.jp; 3Department of Biochemistry and Genetics, Graduate School of Medicine, Chiba University, Chiba 260-8670, Japan; moriya.shogo@chiba-u.jp

**Keywords:** intrahepatic cholangiocarcinoma, *miR-29a-3p*, *miR-29b-3p*, *miR-29c-3p*, tumor-suppressor, *ITGA6*, *ITGB1*, *SP1*

## Abstract

**Simple Summary:**

Even today, there are no effective targeted therapies for intrahepatic cholangiocarcinoma (ICC) patients. Clarifying the molecular pathogenesis of ICC will contribute to the development of treatment strategies for this disease. In this study, we searched for the role of the *miR-29-3p-*family and its association with oncogenic pathway. Interestingly, aberrant expression of *ITGA6* and *ITGB1* was directly regulated by the *miR-29-3p-*family which are involved in multiple oncogenic pathways in ICC, and enhanced malignant transformation of ICC cells. Furthermore, SP1 which is a transcriptional activator of *ITGA6/ITGB1*, is regulated by the *miR-29-3p*-family. These molecules may be novel therapeutic targets for ICC.

**Abstract:**

The aggressive nature of intrahepatic cholangiocarcinoma (ICC) renders it a particularly lethal solid tumor. Searching for therapeutic targets for ICC is an essential challenge in the development of an effective treatment strategy. Our previous studies showed that the *miR-29-3p*-family members (*miR-29a-3p*, *miR-29b-3p* and *miR-29c-3p*) are key tumor-suppressive microRNAs that control many oncogenic genes/pathways in several cancers. In this study, we searched for therapeutic targets for ICC using the *miR-29-3p*-family as a starting point. Our functional studies of cell proliferation, migration and invasion confirmed that the *miR-29-3p*-family act as tumor-suppressors in ICC cells. Moreover, in silico analysis revealed that “focal adhesion”, “ECM-receptor”, “endocytosis”, “PI3K-Akt signaling” and “Hippo signaling” were involved in oncogenic pathways in ICC cells. Our analysis focused on the genes for integrin-α6 (*ITGA6*) and integrin-β1 (*ITGB1*), which are involved in multiple pathways. Overexpression of *ITGA6* and *ITGB1* enhanced malignant transformation of ICC cells. Both *ITGA6* and *ITGB1* were directly regulated by the *miR-29-3p*-family in ICC cells. Interestingly, expression of *ITGA6*/*ITGB1* was positively controlled by the transcription factor SP1, and *SP1* was negatively controlled by the *miR-29-3p*-family. Downregulation of the *miR-29-3p*-family enhanced *SP1*-mediated *ITGA6*/*ITGB1* expression in ICC cells. MicroRNA-based exploration is an attractive strategy for identifying therapeutic targets for ICC.

## 1. Introduction

Cholangiocarcinoma is anatomically divided into intrahepatic cholangiocarcinoma (ICC) and extrahepatic cholangiocarcinoma. From cancer genomic analysis, each cancer contains characteristic gene mutations, suggesting molecular biological differences [1,2,3]. ICC is the second most common cancer of the liver after hepatocellular carcinoma (HCC). ICC constitutes about 5% of primary liver cancers, and global morbidity and mortality of the disease have increased significantly in recent years [4,5,6,7]. Due to the aggressive features of ICC (the high rate of metastasis), the prognosis of patients is very poor (5-year survival rate is less than 10%) [8]. Although surgical resection is the only strategy aiming at a complete cure, only about 30% of patients are operable [9]. Even today, there are no effective anti-cancer drugs or molecularly targeted therapies for ICC patients in the advanced stage of disease [1,2]. The lack of effective diagnostic markers for ICC is a major factor that delays the detection of this disease [10,11]. Clarifying the molecular pathogenesis of ICC based on the latest genomic analyses will contribute to the development of treatment strategies for this disease.

The Human Genome Project has revealed the presence of a vast number of non-coding RNA molecules (ncRNAs) in the human genome. These ncRNAs have important function within cells [12]. MicroRNAs (miRNAs) constitute a set of small ncRNAs (19 to 23 bases) that negatively regulate the expression of RNA transcripts in a sequence-dependent manner [13]. Over the last decade, accumulating evidence has shown that aberrantly expressed miRNAs are closely involved in cancer cell development, metastasis and drug-resistance [12,14,15]. For example, dysregulated miRNAs can disrupt tightly controlled RNA networks, triggering the transition of normal cells to diseased cells.

Identification of aberrantly expressed miRNAs in each type of cancer is the first step towards elucidating the roles of miRNA in the molecular pathogenesis of cancers. Based on miRNA expression signatures, we have revealed some of the roles of tumor-suppressive miRNAs and their controlled oncogenic targets and pathways in gastrointestinal cancers, e.g., esophageal cancer, gastric cancer and pancreatic ductal adenocarcinoma [16,17,18].

Previous study of ICC signatures showed that a total of 30 miRNAs (10 of which were upregulated and 20 were downregulated) were dysregulated in ICC tissues [19]. Moreover, the expression of three miRNAs (*miR-675-5p*, *miR-652-3p* and *miR-338-3p*) was characteristic of ICC. Those miRNA signatures were independent prognostic indicators based on multivariate analysis [19]. Other signatures showed that 38 miRNAs were differentially expressed in tumor and normal tissues [20]. Among these miRNAs, exogenous expression of *miR-320* and *miR-204* negatively controlled *Mcl-1* and *BCl-2* expression, respectively. Expression of those miRNAs facilitates chemotherapeutic drug-triggered apoptosis [20]. Upregulation of *miR-191* in ICC tissues was detected by next-generation sequencing technology [21]. Overexpression of *miR-191* enhanced ICC aggressive phenotypes, e.g., proliferation, invasion and migration abilities in vitro and in vivo [21].

Analysis of our RNA-sequence-based signatures revealed that the *miR-29-3p*-family (*miR-29a-3p*, *miR-29b-3p* and *miR-29c-3p*) was frequently downregulated in several cancers, suggesting these miRNAs acted as pivotal tumor-suppressors in cancer cells [22,23]. Importantly the *miR-29*-*3p-*family is downregulated in ICC tissues [24]. The aim of the study was to investigate the functional significance of the *miR-29-3p*-family and to identify oncogenic targets/pathways in ICC cells. Our ectopic expression assays showed that all members of the *miR-29-3p*-family acted as tumor-suppressors in ICC cells. Several oncogenic pathways (e.g., focal adhesion, ECM-receptor, endocytosis, PI3K–Akt signaling and Hippo signaling) were subject to control by the *miR-29*-*3p-*family. Furthermore, we revealed that the genes for integrin alfa-6 (*ITGA6*) and beta-1(*ITGB1*) were directly regulated by the *miR-29-3p*-family, and their overexpression enhanced migration and invasive abilities in ICC cells. In addition, we showed that transcription factor SP1 is involved in *ITGA6* and *ITGB1* expression, and that *SP1* expression was negatively controlled by the *miR-29*-family in ICC cells.

Downregulation of the *miR-29-3p*-family activated several oncogenic pathways, and these events were closely involved in ICC oncogenesis. Oncogenic signaling pathways mediated by *ITGA6*/*ITGB1* are likely therapeutic targets for this disease. Our miRNA-based approach will accelerate the understanding of the molecular pathogenesis of ICC.

## 2. Results

### 2.1. Downregulation of miR-29a-3p, miR-29b-3p and miR-29c-3p in ICC Clinical Specimens

The expression levels of *miR-29a-3p*, *miR-29b-3p* and *miR-29c-3p* were investigated to utilize the Gene Expression Omnibus (GEO) database (GSE: 53870). These datasets contained 63 ICC tissues and 9 normal intrahepatic bile ducts tissues. Downregulation of *miR-29a-3p*, *miR-29b-3p* and *miR-29c-3p* was confirmed in ICC tissues compared with normal tissues (*p* = 0.0003, *p* < 0.001 and *p* = 0.0210, respectively; Figure 1A). These miRNAs were clustered at two different human loci: *miR-29b-1* and *miR-29a* were at 7q32.3 and *miR-29b-2*, and *miR-29c* was at 1q32.2. Mature sequences of *miR-29a-3p* and *miR-29c-3p* are identical. On the other hand, the mature sequence of *miR-29b-3p* differs at two bases on the 3′end. The seed sequences of the three miRNAs are identical (red letters; Figure 1B).

### 2.2. Effects of Ectopic Expression of miR-29a-3p, miR-29b-3p and miR-29c-3p on ICC Cell Proliferation, Migration and Invasion

The tumor-suppressive roles of *miR-29a-3p*, *miR-29b-3p* and *miR-29c-3p* were assessed by using ectopic expression assays in HuCCT1 and RBE cells. Cell proliferation assays showed no significant effects of these miRNAs transfected in two cell lines (Figure 2A). In contrast, cell migration and invasive abilities were significantly inhibited in *miR-29a-3p*, *miR-29b-3p* and *miR-29c-3p*-transfected cells (HuCCT1 and RBE) compared with mock- or miR-control-transfected cells (Figure 2B,C).

### 2.3. Identification of Putative Pathways and Targets of the miR-29-3p-Family (miR-29a/b/c-3p) Regulation in ICC Cells

The seed sequences of the *miR-29-3p-*family are identical (Figure 1B). Thus, we predicted that the pathways and targets controlled by these miRNAs would be the same. Our strategy for the selection of *miR-29-3p*-family controlled pathways and target genes is shown in Figure 3.

First, based on the TargetScan database (release 7.2), we searched for putative genes that have target sites for the *miR-29-3p-*family (AGCACCA). A total of 3543 genes were identified. Second, we selected the upregulated genes in cholangiocarcinoma tissues compared with normal tissues by using TCGA database (36 cancer and 9 normal tissues, TCGA-CHOL) and Subio platform. A total of 4999 upregulated genes (log_2_ ratio > 1) were identified. Next, the two datasets were merged, and a total of 888 genes were selected as putative targets of *miR-29-3p* (i.e., upregulated genes in cholangiocarcinoma tissues that have *miR-29-3p* binding sites). Finally, we classified 888 genes (Appendix A) based on their molecular functions using KEGG pathways (GeneCodis 4.0).

A total 20 pathways were identified as the *miR-29-3p*-family controlled oncogenic pathways in ICC cells (Table 1; Appendix A). In this study, we focused on *ITGA6* and *ITGB1*, both of which involve multiple pathways, indicating these two genes closely contributed to ICC oncogenesis.

### 2.4. Expression of ITGA6/ITGA6 and ITGB1/ITGB1 Genes and Proteins in ICC Clinical Specimens

We analyzed expression levels of *ITGA6* and *ITGB1* (RNA-Sequence data in 36 cases of TCGA-CHOL) using the GEPIA2 database platform. Expression of both *ITGA6* and *ITGB1* was significantly upregulated (*p* < 0.01; Figure 4A). We also examined the protein expression levels of *ITGA6* and *ITGB1* in an ICC clinical specimen using immunohistochemical methods. Expression of *ITGA6* and *ITGB1* was detected in the cancer cells (Figure 4B–D). Immunostaining confirmed that ITGB1 was expressed in normal cholangiocytes and hepatocytes (Appendix A). Compared with ITGB1, expression of ITGA6 was hardly observed (Appendix A).

### 2.5. Direct Regulation of ITGA6 and ITGB1 by miR-29a-3p, miR-29b-3p and miR-29c-3p in ICC Cells

We investigated direct regulation of *ITGA6* by the *miR-29-3p*-family in ICC cells. Expression levels of both mRNA and protein were significantly reduced by ectopic expression of *miR-29a-3p*, *miR-29b-3p* and *miR-29c-3p* in ICC cells (HuCCT1 and RBE, Figure 5, Appendix A). Similarly, ectopic expression of *miR-29a-3p*, *miR-29b-3p* and *miR-29c-3p* reduced expression levels of *ITGB1*/ITGB1 (Figure 6, Appendix A).

Next, to assess whether the *miR-29-*family directly binds to the *ITGA6* target site in ICC cells, we performed dual-luciferase reporter assays. Luciferase activity was significantly decreased following co-transfection of *miR-29c-3p* and a vector carrying the wild-type *miR-29c-3p* target site. In contrast, luciferase activity was not changed following co-transfection of *miR-29c-3p* and a vector carrying the deletion-type of the *miR-29c-3p* target site (Figure 7A). These results indicated that *ITGA6* was directly regulated by *miR-29a-3p*, *miR-29b-3p* and *miR-29c-3p* in ICC cells. Similar results were confirmed in the dual-luciferase reporter assays of *ITGB1* and *miR-29c-3p*, indicating that *ITGB1* was directly regulated by *miR-29a-3p*, *miR-29b-3p* and *miR-29c-3p* in ICC cells (Figure 7B).

### 2.6. Effects of Knockdown of ITGA6 and ITGB1 on Cell Proliferation, Migration, and Invasion in ICC Cells

To assess the oncogenic functions of *ITGA6/ITGB1* in ICC cells, we conducted knockdown assays of corresponding genes using small interfering RNAs (siRNAs). First, we evaluated the mRNA and protein knockdown efficiencies of siRNAs that targeted *ITGA6* and *ITGB1* in HuCCT1 and RBE cells. The expression levels of *ITGA6*/ITGA6 and *ITGB1*/ITGB1 were significantly reduced by all siRNA-transfected cells, HuCCT1 and RBE (Figure 8 and Figure 9, Appendix A). Functional assays using siRNAs demonstrated that knockdown of ITGA6 and ITGB1 suppressed cancer cell malignant phenotypes, especially cell migration and invasive abilities (Figure 10 and Figure 11). Furthermore, both integrins (*ITGA6* and *ITGB1*) were knocked down at the same time to examine the tumor suppressor effects, e.g., cell proliferation and migration, and invasive abilities. By knocking down both integrins, the malignant phenotypes of cancer cells were further controlled (Appendix A).

### 2.7. Identification of Transcriptional Regulators of ITGA6 and ITGB1 in ICC Cells

We further investigated the involvement of transcription factors (TFs) that positively regulated *ITGA6* and *ITGB1* expression in ICC cells based on in silico database analysis. GeneCodis database analysis and previous studies identified a total of 18 TFs (Table 2) [25,26,27].

Among these TFs, expression of 12 TFs was upregulated in cholangiocarcinoma tissues (Appendix A). Moreover, expressions of six TF genes (*SP1*, *CREB1*, *MAX*, *FHL2*, *RFX1*, and *HIF1A*) was positively correlated with the expression of *ITGA6* and *ITGB1* in cholangiocarcinoma tissues (Appendix A). These TFs might be involved with the enhanced expression of *ITGA6* and *ITGB1* in ICC cells. Thus, further investigation was conducted.

### 2.8. Regulation of SP1 by miR-29a-3p, miR-29b-3p and miR-29c-3p in ICC Cells

Expression levels of *SP1*/SP1 in clinical specimens and positive correlations of *SP1*/*ITGA6* and *SP1*/*ITGB1* are shown in Figure 12 and Appendix A.

Interestingly, *miR-29-3p* binding sites were detected in the 3′-UTR region of *SP1* mRNA. To evaluate whether direct regulation of *SP1* by *miR-29-3p-family* in ICC cells occurred, we performed dual-luciferase reporter assays. Luciferase activity was significantly decreased following co-transfection of *miR-29c-3p* and a vector carrying the wild-type *miR-29c-3p* target site. In contrast, luciferase activity was not changed following co-transfection of *miR-29c-3p* and a vector carrying the deletion-type of the *miR-29c-3p* tar-get site (Figure 13A). These results indicated that *SP1* was directly regulated by the *miR-29-3p-*family in ICC cells. Furthermore, both mRNA and protein expression levels of *SP1*/SP1 were reduced by ectopic expression of *miR-29a-3p*, *miR-29b-3p* and *miR-29c-3p* in HuCCT1 and RBE cells (Figure 13B,C, Appendix A). These data indicated that downregulation of *miR-29a-3p*, *miR-29b-3p* and *miR-29c-3p* enhanced *SP1* expression, and this event accelerated the overexpression of *ITGA6* and *ITGB1* in ICC cells.

## 3. Discussion

Among primary liver cancers, the incidence of ICC (15% of liver cancers) is second only to hepatocellular carcinoma (HCC), and the global incidence of ICC has increased over the past three decades [4,8]. Patients with ICC are advanced at the time of initial diagnosis because there are no effective early diagnostic markers [10,11]. Unfortunately, there is no effective treatment for inoperable cases [1,2]. Many studies have attempted to elucidate the molecular mechanisms underlying ICC [28,29]. Recently, genome-based analyses, including ncRNAs and miRNAs, have been vigorously conducted [12,15].

Recent study demonstrated that lncRNA-PAICC was overexpressed in ICC tissues, and expression of PAICC enhanced proliferation and invasion of ICC cells [30]. Importantly, oncogenic PAICC functioned as competitive endogenous RNA (ceRNA) that adsorbed tumor-suppressive miRNAs, *miR-141-3p* and *miR-27a-3p*, in cancer cells. Silencing of *miR-141-3p* and *miR-27a-3p* induced activation of the Hippo pathway in ICC cells [30]. Similarly, overexpression of lncRNA-UCA1 sponged *miR-122*, and this event enhanced proliferation and invasion of ICC cells [31]. Aberrant expression of lncRNAs and silencing tumor-suppressive miRNAs was closely associated with ICC oncogenesis [31].

In this study, we focused on the *miR-29-3p*-family because previous ICC signature and our miRNA signatures in several cancers revealed that the *miR-29-3p*-family was significantly downregulated in cancer tissues, suggesting that its members acted as key regulators, negatively controlling pivotal oncogenic targets and pathways [22,23,32]. Downregulation and tumor-suppressive functions of the *miR-29-3p*-family have been reported in several types of cancers [33,34]. Thus far, no detailed functional and targeted analysis of the *miR-29-3p*-family in ICC cells has been performed. In HCC cells, several studies showed that the *miR-29-3p*-family acted as tumor-suppressive miRNAs via negative control of several oncogenic targets, e.g., *IGF2BP1*, *RPS15A* and *LOXL2* [35,36,37]. In cholangiocarcinoma, the serum concentration of *miR-29b-3p* was elevated compared to healthy controls and patients with primary sclerosing cholangitis [38]. The molecular mechanism increasing the level of *miR-29b-3p* in serum is not understood. It may be released from cancerous tissue.

Our next interest was the search for oncogenic molecular networks in ICC cells that were controlled by the tumor-suppressive *miR-29**-3p*-family. Our in silico analysis identified 20 pathways, including “focal adhesion”, “ECM-receptor”, “endocytosis”, “PI3K-Akt signaling” and “Hippo signaling”. Analysis of the genes contained in these molecular pathways revealed that *ITGA6* and *ITGB1* were involved in multiple pathways. Integrins are cell surface proteins that interact with the extracellular matrix (ECM), modulating cell characteristics, such as cell shape, proliferation, and motility [39]. We also analyzed expression levels of other ITGs that form heterodimers with *ITGB1* in ICC clinical specimens using GEPIA2 database (Appendix A). The expression of some ITGs, e.g., *ITGAV*, *ITGA2*, *ITGA3*, *ITGA5*, and *ITGA7* were upregulated in ICC tissues. Additionally, upregulation of *ITGB4* (form heterodimers with *ITGA6*) was detected in ICC tissues. It was suggested that not only the *ITGA6*/*ITGB1* dimer but also the ITGs that form dimers with each of these may play an important role in ICC. Additionally, liver fibrosis is characterized by an abnormal accumulation of extracellular matrix (ECM) and is a common feature of chronic liver damage. The hepatic stellate cell (HSC) is involved in liver fibrosis, which is the formation of scar tissue in response to liver damage. Previous study showed that *miR-29b* was downregulated during HSC activation [40]. Ectopic expression of *miR-29b* inhibited several genes involved in HSC activation, e.g., *COL1A1*, *COL1A2*, *DDR2*, *FN1*, and *ITGB1* [41]. A vast number of studies have shown that aberrant expression of ECM/integrin-mediated oncogenic signaling enhanced cancer cell malignant transformation, e.g., invasion, migration, the epithelial mesenchymal transition, metastasis and drug resistance [17,32].

Previous studies showed that several miRNAs (e.g., *miR-29*, *miR-124-3p*, *miR-150*, *miR-218* and *miR-199*) directly controlled expression of *ITGA6* and *ITGB1* in several cancers. Ectopic expression of these miRNAs attenuated cancer cell migration, invasion and metastasis [17,32,42,43]. Our previous study of pancreatic ductal adenocarcinoma (PDAC) cells showed that aberrant expression of *ITGA3* and *ITGB1* (genes for two integrins that form homodimers) was significantly associated with poor prognosis of patients with PDAC [17]. Expression of *ITGA3* and *ITGB1* was directly regulated by tumor-suppressive *miR-124-3p*. Moreover, overexpression of *miR-124-3p* inhibited ITGA3/ITGB1-mediated oncogenic signaling, and attenuated cancer cell migration and invasive abilities [17]. In head and neck squamous cell carcinoma cells, ectopic expression of the *miR-29-3p*-family inhibited ITGB1-mediated oncogenic signaling [22]. At this time, no molecular-targeted drugs targeting the integrin-family are used for ICC. Searching for specific integrin-mediated molecular pathways in ICC will help develop new therapeutic agents. Integrin inhibitors have been clinically applied in autoimmune diseases, thrombosis, and several cancers, but with poor results, especially for malignant tumors [44,45,46]. This may be due to two aspects of their function: (1) integrins require structural changes due to their inside-out signals, and (2) multiple integrin units’ function in cancer cells [47]. Therefore, future development of therapy must control both integrin-related cascades as well as the presence of currently expressed ligands. In this study, we also investigated the involvement of transcription factors (TFs) that positively regulated *ITGA6* and *ITGB1* expression in ICC cells. Our in silico analysis revealed that a total of six TFs (*SP1*, *CREB1*, *MAX*, *FHL2*, *RFX1* and *HIF1A*) were upregulated in cholangiocarcinoma tissues. We also found a positive correlation between *ITGA6* and *ITGB1* expression. Notably, among these TFs, *SP1* has a *miR-29*-family binding site in the 3′-UTR region. Several studies showed that *miR-29b* and *miR-29c* directly regulated *SP1* [48,49]. Our present data also showed inhibited expression of *SP1* by ectopic expression of the *miR-29-3p*-family in ICC cells.

## 4. Conclusions

These findings indicated that downregulation of the *miR-29-3p*-family caused upregulation of *ITGA6*/*ITGB1*, and their transcriptional modulator *SP1* in ICC cells. Overexpression of *SP1* induces the expression of several genes involved in malignant transformation of cancer cells [50]. Therefore, reducing the expression of *SP1* should permit control of cancers. A vast number of studies indicated that small-molecule drugs and natural products (bortezomib, retinoids, aspirin, metformin, and curcumin) downregulated *SP1* expression in cancer cells [51,52]. It may be possible to control ICC by combining it with existing anti-cancer drugs.

## 5. Materials and Methods

### 5.1. ICC Cell Lines and Cell Culture

We used 2 human ICC cell lines: HuCCT1 and RBE, both purchased from the RIKEN Cell Bank (Tsukuba, Ibaraki, Japan). The cell lines were maintained in RPMI 1640 medium supplemented with 10% fetal bovine serum in a humidified atmosphere of 5% CO_2_ and 95% air at 37 °C. These cell lines were transferred from the RIKEN Bank in 2020 and were used in the present analysis.

### 5.2. RNA Extraction and Quantitative Real-Time Reverse Transcription Polymerase Chain Reaction (qRT-PCR)

The methods for total RNA extraction from clinical specimens and cell lines and for qRT-PCR have been described previously [16,17]. *GUSB* was used as the normalized control. The reagents used in the analysis are listed in Appendix A.

### 5.3. Transfection of miRNAs, siRNAs, and Plasmid Vectors into ICC Cells

The transfection procedures of miRNAs, siRNAs, and plasmid vectors into ICC cells were described previously [16,17]. The reagents used in this study are listed in Appendix A.

### 5.4. Functional Assays in ICC Cells (Cell Proliferation, Migration and Invasion)

The procedures for conducting the functional assays in cancer cells (e.g., proliferation, migration and invasion) have been described previously [16,17]. In brief, for proliferation assays, cells were transferred to 96-well plates. HuCCT1 and RBE cells were plated at 2.4 × 10^3^ cells per well. Cell proliferation was evaluated using the XTT assays 72 h after the transfection procedure. For migration and invasion assays, HuCCT1 cells or RBE cells at 1.2 × 10^5^ were transfected in 6-well plates. After 72 h, HuCCT1 or RBE cells were added into each chamber at 2.5 × 10^5^ per well. After 48 h, the cells on the lower surface were counted for analysis. All experiments were performed in triplicate.

### 5.5. Expression Analysis of miRNA, Target Genes and Transcriptional Factors Using Public Databases

To search for downregulated miRNAs in ICC clinical specimens, we obtained expression data from the GEO database (GSE53870). The expression data of 63 patients with ICC and 9 normal intrahepatic bile ducts are stored in the dataset. We imported TCGA-CHOL RNA seq data into Subio platform (https://www.subioplatform.com/ (accessed on 29 October 2020)) and analyzed gene expression by using the “compare 2 groups” tool of Subio platform. The expression level of each gene was extracted and analyzed from the TCGA-CHOL database through the GEPIA2 platform (http://gepia2.cancer-pku.cn/#index (accessed on 10 February 2021)).

### 5.6. Plasmid Construction and Dual-Luciferase Reporter Assays

Plasmid vectors containing wild type and deletion type sequences of *miR-29-3p* binding site in 3′-UTR of *ITGA6* or *ITGB1* or *SP1* were prepared. The predicted binding site sequences were obtained from the TargetScanHuman database (https://www.targetscan.org/, release 7.2 (accessed on 1 December 2020)).

Cells were co-transfected with *miR-29c-3p* and the plasmid vectors for 36 h. The detailed procedures for dual-luciferase reporter assay were described in our previous studies [16,17].

### 5.7. Western Blotting and Immunohistochemistry

Cell lysates were prepared in RIPA buffer 72 h after transfection. Polyacrylamide gel electrophoresis was performed using 30 μg of protein lysate, and the protein lysates were transferred onto PVDF membranes (Thermo Fisher Scientific, Waltham, MA, USA). Membranes were blocked with skim milk and incubated with the indicated primary antibodies overnight at 4 °C. The antibodies used in this study are shown in Appendix A. GAPDH was used as the internal control. The quantitation of the protein band intensity was analyzed by ImageJ software (NIH, Bethesda, MD, USA). We assessed expression of ITGA6, ITGB1 and SP1 proteins by immunohistochemistry. The procedure for immunostaining was described previously [16,17]. Clinical sample were obtained from a patient following resection at Kagoshima University Hospital at March 2017. The patient was diagnosed with mass-forming type ICC, T3N0M0 stage IIIA according to the 8th edition of the Union for International Cancer Control (UICC).

### 5.8. Statistical Analyses

All statistical analyses were performed using JMP Pro 15 (SAS Institute Inc., Cary, NC, USA). Differences between 2 groups were evaluated using Mann–Whitney U tests. For multiple groups, one-way analysis of variance and Dunnett’s test were applied. Correlation coefficients were evaluated using Pearson’s test. All data are presented as the mean ± SD. *p*-values less than 0.05 were considered significant.

## Figures and Tables

**Figure 1 cancers-13-02804-f001:**
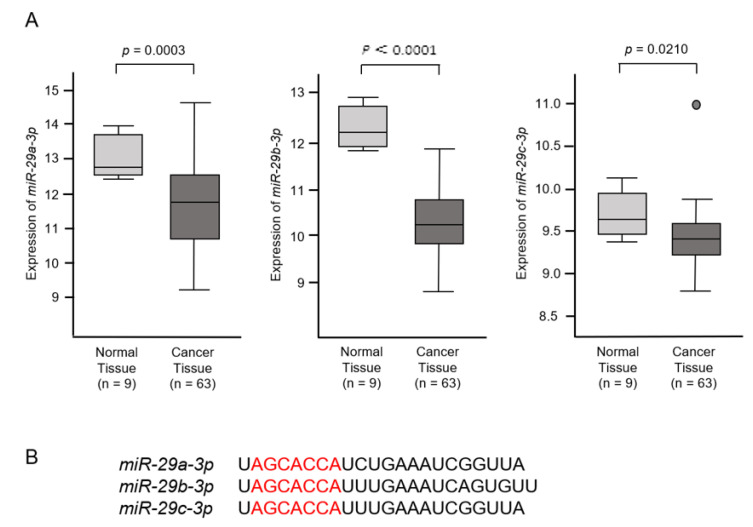
In silico expression analysis of *miR-29a-3p*, *miR-29b-3p* and *miR-29c-3p*. (**A**) Expression levels of *miR-29a-3p*, *miR-29b-3p* and *miR-29c-3p* in ICC tissues. A total of 63 ICC cancer tissues and 9 normal intrahepatic bile ducts (NIBDs) were analyzed. (**B**) Mature sequences of the miR-29-family are shown. Mature sequences of *miR-29a-3p* and *miR-29c-3p* are identical. The mature sequence of *miR-29b-3p* differs in 2 bases on the 3′ end. The seed sequences of the 3 miRNAs are identical (red letters).

**Figure 2 cancers-13-02804-f002:**
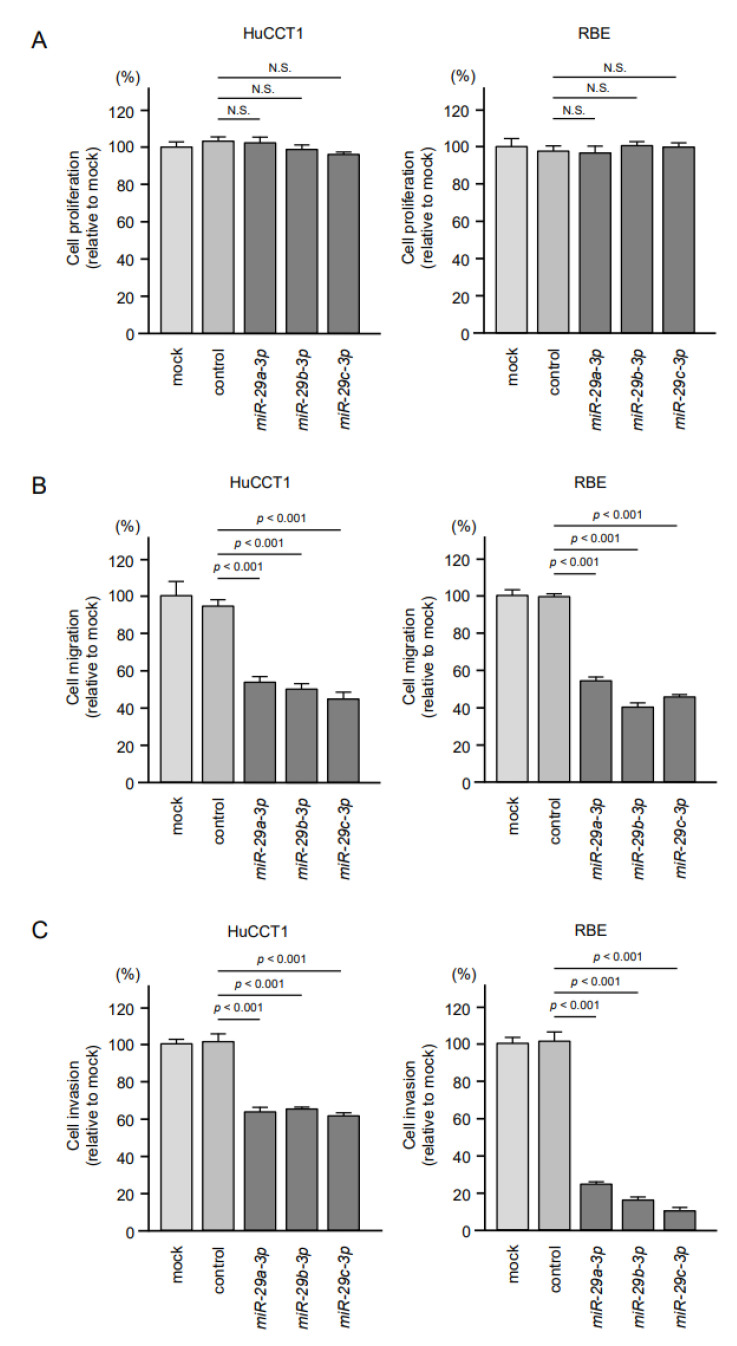
Effects of *miR-29a-3p*, *miR-29b-3p* and *miR-29c-3p* on functions (cell proliferation, migration and invasion) of intrahepatic cholangiocarcinoma (ICC) cell lines (HuCCT1 and RBE). (**A**) Cell proliferation was assessed using XTT assays. Data were collected 72 h after miicroRNA transfection. Ectopic expression of *miR-29a-3p*, *miR-29b-3p* and *miR-29c-3p* did not affect cell proliferation. (**B**,**C**) Cell migration and invasive abilities were significantly blocked by ectopic expression of *miR-29a-3p*, *miR-29b-3p* and *miR-29c-3p* in ICC cells. Error bars represent mean ± standard error (SE).

**Figure 3 cancers-13-02804-f003:**
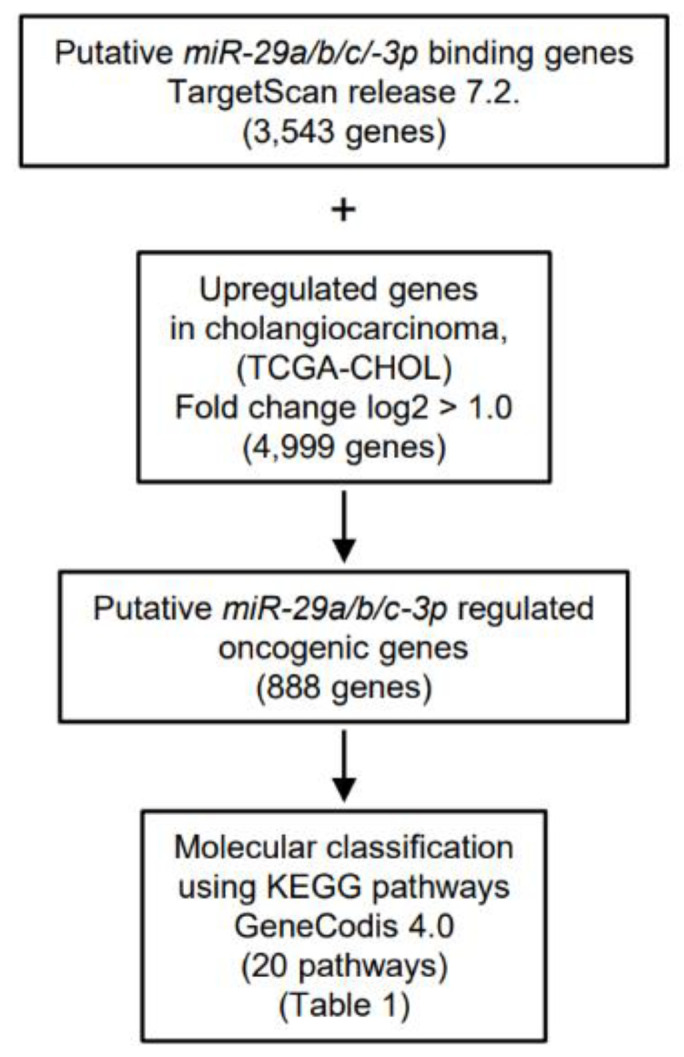
Flowchart summarizing the search for oncogenic targets of *miR-29-3p* regulation in ICC cells. To identify *miR-29-3p* target genes, we screened putative targets using the TargetScan database and TCGA database. A total of 888 targets were upregulated in cholangiocarcinoma tissues. These genes were then categorized into KEGG (Kyoto Encyclopedia of Genes and Genomes) pathways using the GeneCodis database. Finally, 20 pathways that were regulated by *miR-29-3p* were identified.

**Figure 4 cancers-13-02804-f004:**
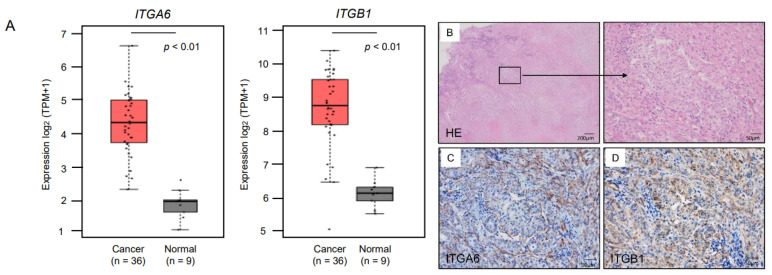
Expression of *ITGA6* and *ITGB1* in ICC. (**A**) Expression levels of *ITGA6* and *ITGB1* in ICC tissues and normal tissues obtained from TCGA-CHOL based on the GEPIA2 platform. (**B**) H&E-stained sections of human ICC tissue. Areas in the boxed region at the left are shown magnified at right. (**C**,**D**) Representative images of tissues immunostained for ITGA6 and ITGB1. ITGA6 stained positive in the membranes of cancer cells in a clinical sample (original magnification × 100). ITGB1 stained positive in the membranes, cytoplasm and a part of the nuclei of cancer cells in a clinical sample (original magnification × 100).

**Figure 5 cancers-13-02804-f005:**
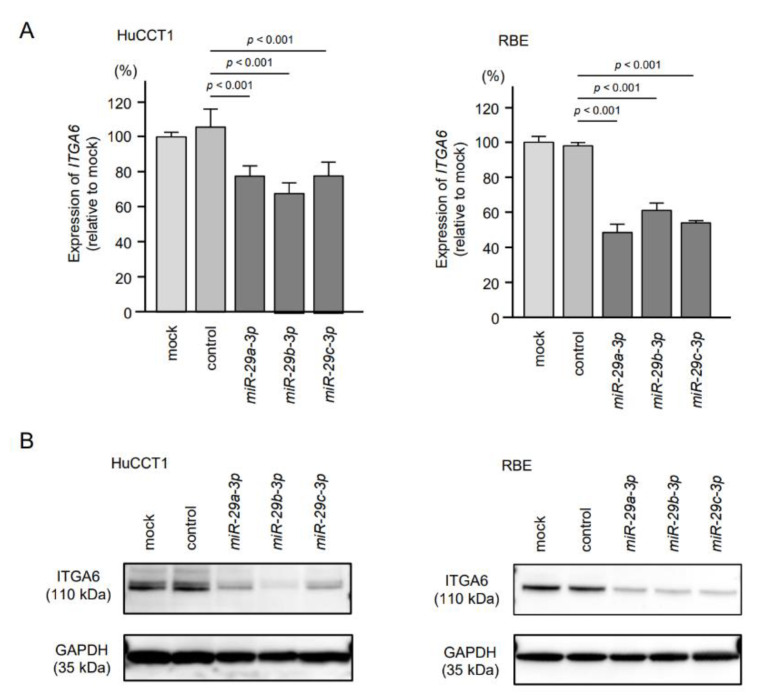
Regulation of *ITGA6*/ITGA6 expression by ectopic expression of *miR-29a-3p*, *miR-29b-3p* and *miR-29c-3p* in intrahepatic cholangiocarcinoma (ICC) cells. (**A**) Expression levels of *ITGA6* were significantly reduced by *miR-29-3p* transfection into ICC cells. *GUSB* was used as an internal control. (**B**) Protein expression levels of ITGA6 were significantly reduced by *miR-29-*family transfection into ICC cells (72 h after transfection). GAPDH was used as an internal control.

**Figure 6 cancers-13-02804-f006:**
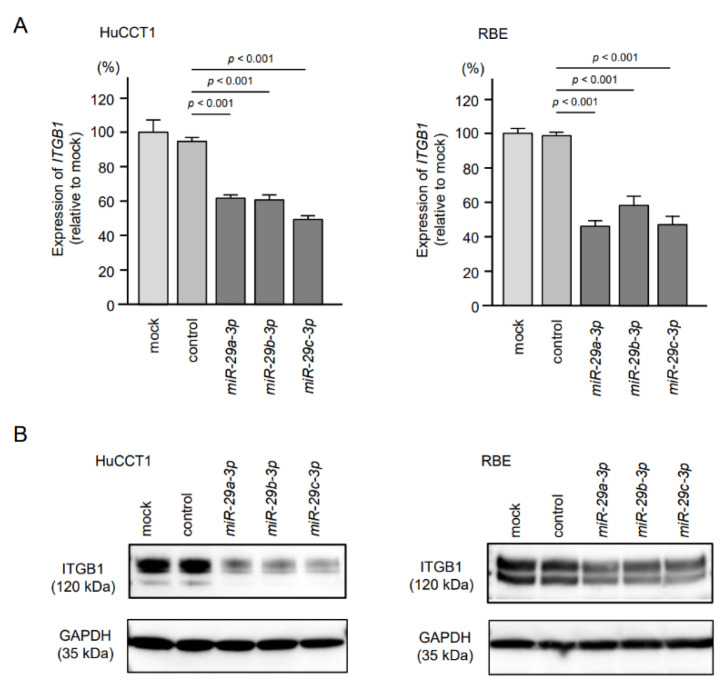
Regulation of *ITGB1*/ITGB1 expression by ectopic expression of *miR-29a-3p*, *miR-29b-3p* and *miR-29c-3p* in intrahepatic cholangiocarcinoma (ICC) cells. (**A**) Expression levels of *ITGB1* were significantly reduced by *miR-29-3p* transfection into ICC cells. *GUSB* was used as an internal control. (**B**) Protein expression levels of ITGB1 were significantly reduced by *miR-29-3p* transfection into ICC cells (72 h after transfection). GAPDH was used as an internal control.

**Figure 7 cancers-13-02804-f007:**
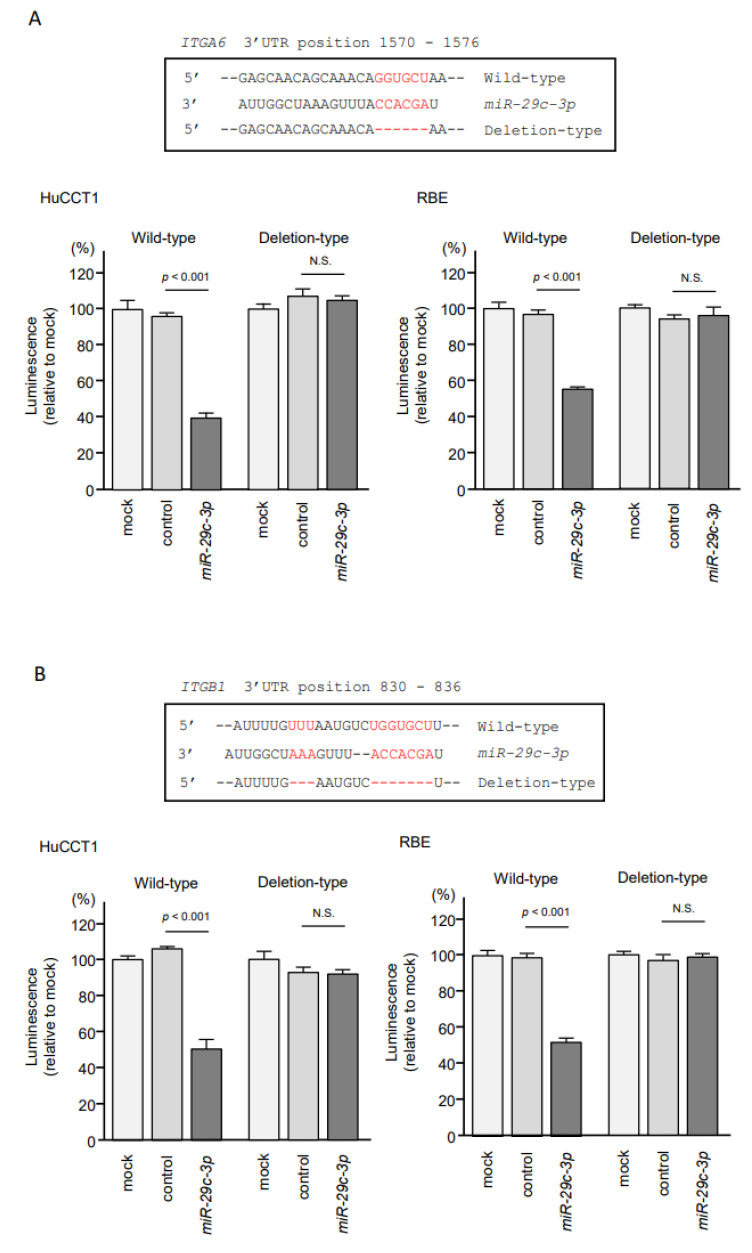
Direct regulation of *ITGA6* and *ITGB1* expression by *miR-29-3p* in ICC cells. (**A**) Dual-luciferase reporter assays showed that luminescence activity was reduced by co-transfection with wild-type vector (containing the *miR-29-3p* binding site of *ITGA6*) and m*iR-29c-3p* precursor in HuCCT1 and RBE cells. In contrast, no change of luminescence activity occurred after co-transfection with deletion-type vector luciferase activity with *miR-29-*3p transfection into ICC cells. Normalized data were calculated as the ratios of *Renilla*/firefly. (**B**) The luminescence activity was reduced by co-transfection with wild-type vector (containing *miR-29-3p* binding site of *ITGB1*) and m*iR-29c-3p* precursor in HuCCT1 and RBE cells. In contrast, no change of luminescence activity occurred after co-transfection with deletion-type vector luciferase activity.

**Figure 8 cancers-13-02804-f008:**
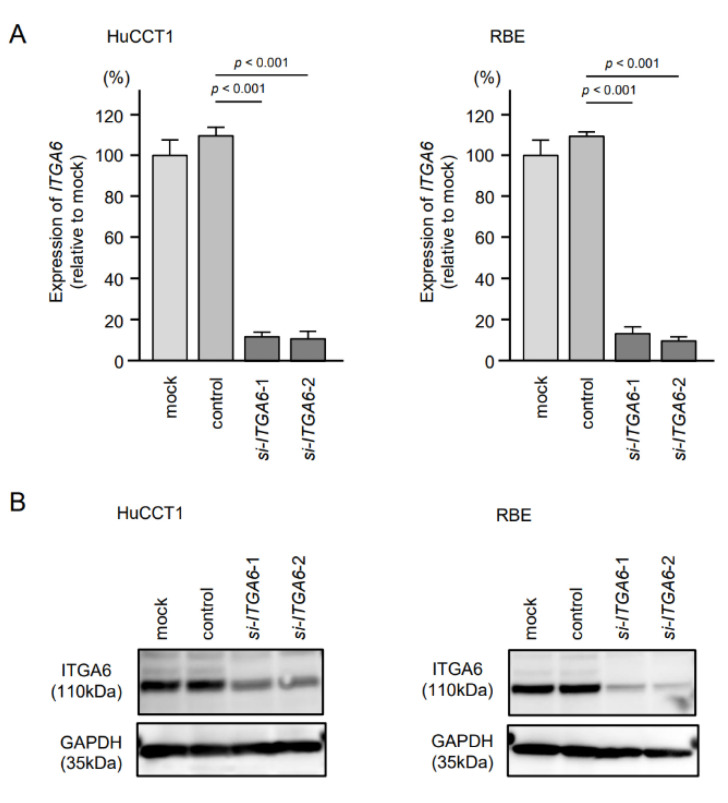
Knockdown efficiency of si*ITGA6* in HuCCt1 and RBE cells. (**A**) Expression of *ITGA6* in ICC cell lines was evaluated 72 h after transfection with *si-ITGA6-1* or *si-ITGA6-2*. *GUSB* was used as an internal control. (**B**) Expression of ITGA6 in ICC cell lines was evaluated by Western blot analysis 72 h after transfection with *si-ITGA6-1* or *si-ITGA6-2*. GAPDH was used as internal control.

**Figure 9 cancers-13-02804-f009:**
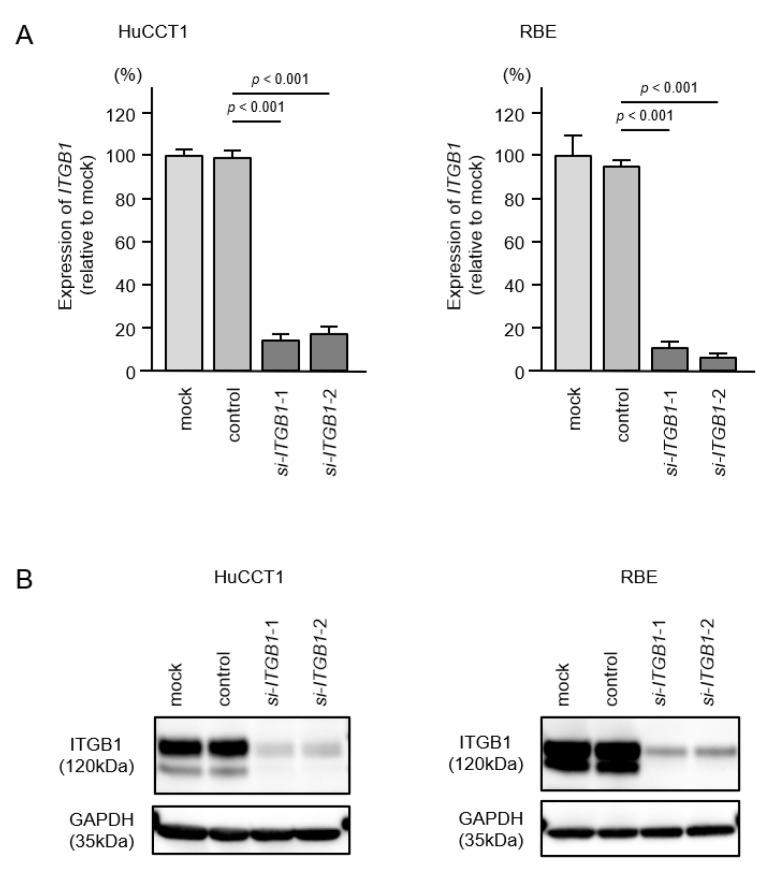
Knockdown efficiency of si*ITGB1* in HuCCT1 and RBE cells. (**A**) Expression of *ITGB1* in ICC cell lines was evaluated 72 h after transfection with *si-ITGB1-1* or *si-ITGB1-2*. *GUSB* was used as an internal control. (**B**) Expression of ITGB1 in intrahepatic cholangiocarcinoma (ICC) cell lines was evaluated by Western blot analysis 72 h after transfection with *si-ITGB1-1* and *si-ITGB1-2*. GAPDH was used as internal control.

**Figure 10 cancers-13-02804-f010:**
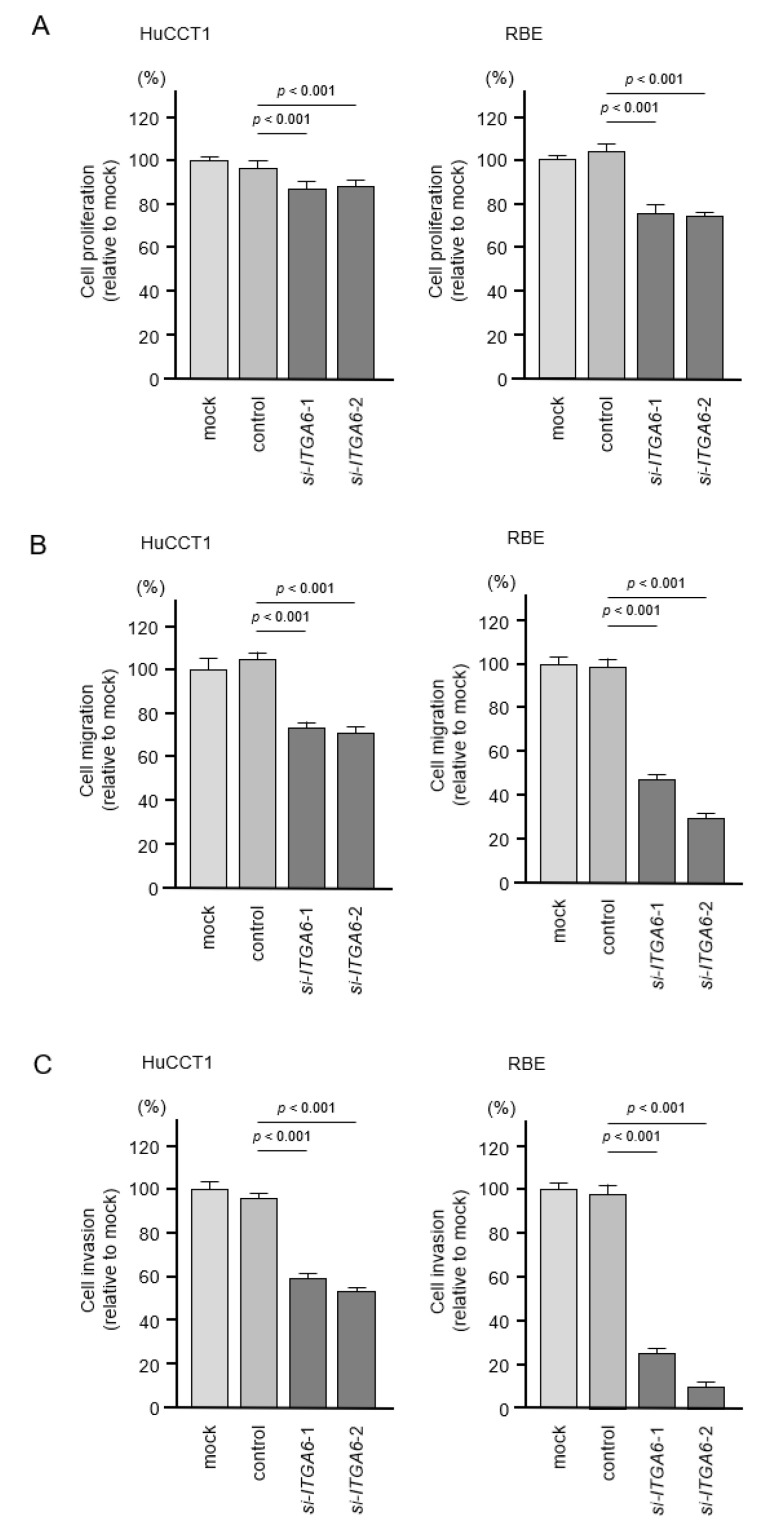
Effects of knockdown of *ITGA6* on cell proliferation, migration and invasion in HuCCT1 and RBE cell lines. (**A**) Cell proliferation was assessed using XTT assays. Data were collected 72 h after miRNA transfection. (**B**) Cell migration was assessed using wound healing assays. (**C**) Cell invasion was determined 72 h after seeding microRNA-transfected cells into chambers using Matrigel invasion assays. Error bars are represented as mean ± standard error (SE).

**Figure 11 cancers-13-02804-f011:**
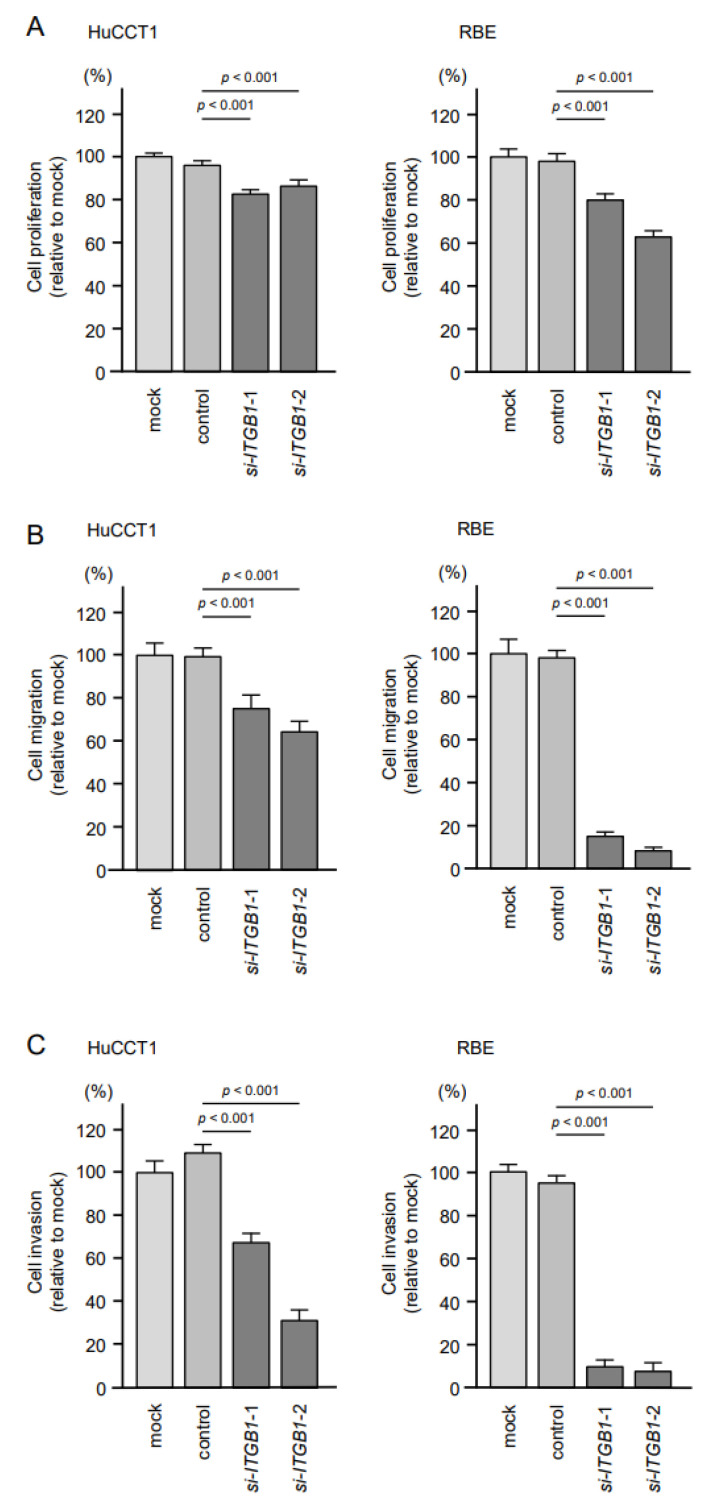
Effects of knockdown of *ITGB1* on cell proliferation, migration and invasion in HuCCT1 and RBE. (**A**) Cell proliferation was assessed using XTT assays. Data were collected 72 h after miRNA transfection. (**B**) Cell migration was assessed using wound healing assays. (**C**) Cell invasion was determined 72 h after seeding miRNA-transfected cells into chambers using Matrigel invasion assays. Error bars are represented as mean ± standard error (SE).

**Figure 12 cancers-13-02804-f012:**
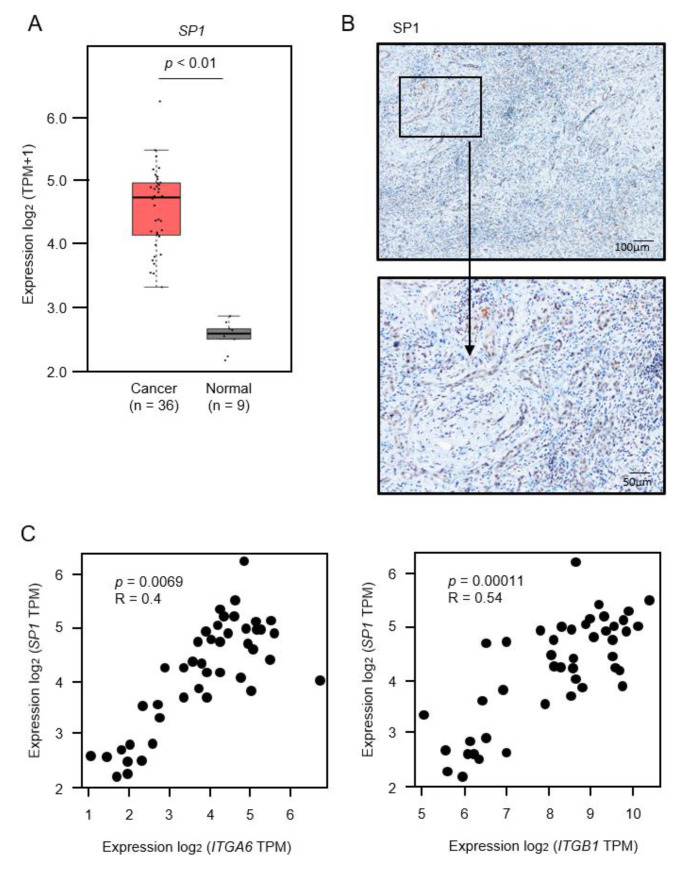
Expression of *SP1* in intrahepatic cholangiocarcinoma (ICC) clinical specimens. (**A**) Expression levels of *SP1* in ICC tissues, normal tissues obtained from TCGA-CHOL through the GEPIA2 platform. (**B**) Immunostaining of SP1 in ICC clinical specimen. Areas in the boxed region above are shown magnified below (original magnification above: ×40, below: ×200). (**C**) Correlation of expression of *SP1*/*ITGA6* and *SP11/ITGB1* in cholangiocarcinoma tissues through GEPIA2 platform. Pearson’s rank tests between the expression levels of *SP1/ITGA6* and *SP1/ITGB1*.

**Figure 13 cancers-13-02804-f013:**
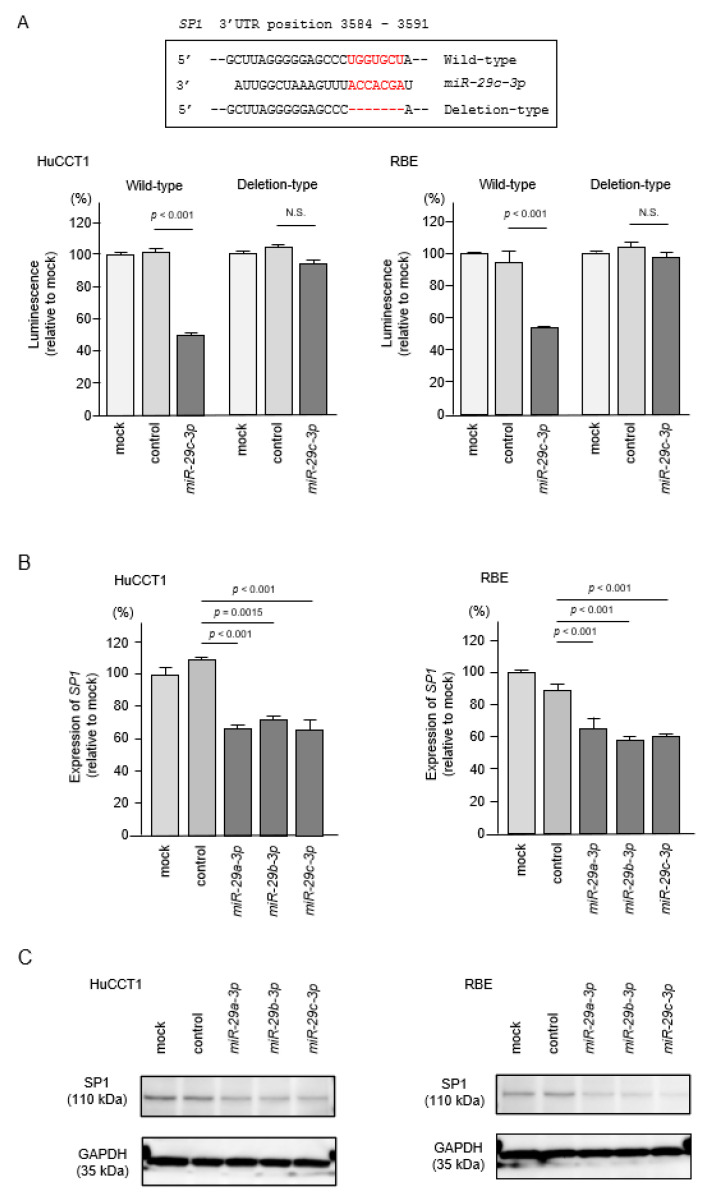
Direct regulation of *SP1* by *miR-29*-family in intrahepatic cholangiocarcinoma (ICC) cells. (**A**) The luminescence activity was reduced by co-transfection with wild-type vector (containing *miR-29-3p* binding site of *SP1*) and m*iR-29c-3p* precursor in HuCCT1 and RBE cells. In contrast, no change of luminescence activity occurred after co-transfection with deletion-type vector luciferase activity. (**B**) Expression levels of *SP1* were significantly reduced by *miR-29a-3p*, *miR-29b-3p* and *miR-29c-3p* transfection into ICC cells. *GUSB* was used as an internal control. (**C**) Protein expression levels of SP1 were significantly reduced by *miR-29a-3p*, *miR-29b-3p*, and *miR-29c-3p* transfection into ICC cells (72 h after transfection). GAPDH was used as an internal control.

**Table 1 cancers-13-02804-t001:** Significantly enriched pathways regulated by the *miR-29-3p-*family in ICC cells.

Number of Genes	*p*-Value	Annotations
30	1.07 × 10^−6^	(KEGG) 04510: Focal adhesion
19	2.04 × 10^−6^	(KEGG) 05222: Small cell lung cancer
18	4.72 × 10^−5^	(KEGG) 04974: Protein digestion and absorption
48	6.98 × 10^−5^	(KEGG) 05200: Pathways in cancer
35	7.63 × 10^−5^	(KEGG) 05165: Human papillomavirus infection
16	8.83 × 10^−5^	(KEGG) 04512: ECM-receptor interaction
28	0.000184427	(KEGG) 04144: Endocytosis
34	0.000560778	(KEGG) 04151: PI3K-Akt signaling pathway
20	0.000585526	(KEGG) 04390: Hippo signaling pathway
13	0.00911125	(KEGG) 05215: Prostate cancer
13	0.0112006	(KEGG) 04933: AGE-RAGE signaling pathway in diabetic complications
12	0.0247066	(KEGG) 05231: Choline metabolism in cancer
18	0.0252777	(KEGG) 04360: Axon guidance
10	0.0253268	(KEGG) 05218: Melanoma
8	0.026522	(KEGG) 04340: Hedgehog signaling pathway
15	0.0283549	(KEGG) 04540: Signaling pathways regulating pluripotency of stem cells
19	0.0302379	(KEGG) 05205: Proteoglycans in cancer
15	0.0354345	(KEGG) 04072: Phospholipase D signaling pathway
11	0.0412279	(KEGG) 04350: TGF-beta signaling pathway
20	0.0463112	(KEGG) 04014: Ras signaling pathway

**Table 2 cancers-13-02804-t002:** Transcription factors for ITGA6 or ITGB1.

Gene Symbol	Gene Name	Expression in ICC ^1^ Cancer Tissues (*p*-Value)	Correlation with *ITGA6* (*p*-Value)	Correlation with *ITGB1 (**p*-Value)	*miR-29-3p* Binding Site	Reference
*SP1*	Sp1 transcription factor	<0.01	<0.01	<0.01	+	[24,25]
*CREB1*	cAMP responsive element binding protein 1	<0.01	<0.01	<0.01	−	[25]
*MAX*	MYC associated factor X	<0.01	<0.01	0.011	−	[25]
*FHL2*	four and a half LIM domains 2	<0.01	<0.01	<0.01	−	[25]
*RFX1*	regulatory factor X, 1 (influences HLA class II expression)	<0.01	<0.01	<0.01	−	[25]
*HIF1A*	hypoxia inducible factor 1, alpha subunit (basic helix-loop-helix transcription factor)	<0.01	<0.01	0.019	−	[25]
*TFAP2A*	transcription factor AP-2 alpha (activating enhancer binding protein 2 alpha)	<0.01	N.S	<0.01	−	[25]
*CUX1*	cut-like homeobox 1	<0.01	<0.01	N.S	−	[25]
*USF1*	upstream transcription factor 1	<0.01	N.S	<0.01	−	[25]
*TFAP2C*	transcription factor AP-2 gamma (activating enhancer binding protein 2 gamma)	<0.01	N.S	N.S	+	[25]
*FOXM1*	forkhead box M1	<0.01	N.S	N.S	−	[26]
*SPI1*	Spi-1 proto-oncogene	<0.05	N.S	N.S	−	[25]
*MXI1*	MAX interactor 1, dimerization protein	N.S ^2^	0.011	N.S	+	[25]
*PAX6*	paired box 6	N.S	<0.01	<0.01	−	[25]
*ESR1*	estrogen receptor 1	N.S	0.015	<0.01	−	[25]
*MEF2A*	myocyte enhancer factor 2A	N.S	<0.01	0.038	−	[25]
*HOXD3*	homeobox D3	N.S	0.012	N.S	−	[25]
*MYC*	MYC proto-oncogene, bHLH transcription factor	N.S	N.S	N.S	−	[25]

^1^ Intrahepatic cholangiocarcinoma, ^2^ No significant difference.

## Data Availability

The data presented in this study are available on request from the corresponding author.

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
