# Peer review of "Molecular Pathogenesis and Regulation of the miR-29-3p-Family: Involvement of ITGA6 and ITGB1 in Intra-Hepatic Cholangiocarcinoma"

_cancers, 2021, doi:10.3390/cancers13112804_

Round 1
Reviewer 1 Report
The manuscript by Hozaka et al investigates the role of IntegrinA6 and B1 in cholangiocarcinoma. The manuscript is clear, albeit the study is quite a simple one. There are a number of questions that I would like to have addressed by the authors:
- In figure 4, what does the protein expression of these two integrins look like in the normal liver? Is it restricted to certain cells (cholangiocytes) or more generally?
- Throughout the study (and particularly in figure 8) its not clear whether B1A6 are co-expressed in the same cells. Can the authors show co-localisation in cells or Co-IP...the assumption is in this paper that these receptors are a heterodimer, but that is not proven in this context.
- What's not clear to me is that in figure 9 whether integrins can affect XTT, how do you know that this isn't mitochondrial dysfunction - are there XTT
- In 10A, the Y axis is labelled with A6, presumably this should be B1?
- The proliferation and migration effects are different between IntergrinA6 and B1, have you evaluated whether the downstream changes are the same with KD of the individual changes?
- What happens if both of the integrins are knocked down at the same time, are the phenotypes additive?
- Can you conform whether SP1 protein is expressed in tumour tissue?
Author Response
Revise letter(Manuscript ID: cancers-1186122)
May 28, 2021
Dr. Takahiro Kodama
Guest Editor
Department of Gastroenterology and Hepatology,
Osaka University Graduate School of Medicine,
Osaka, Japan
Dear Dr. Kodama:
Thank you for your constructive comments and suggestions. We believe that our manuscript has been greatly improved and is now suitable for publication in Cancers. We agree with your and the reviewers’ comments and suggestions. All changes are highlighted in the revised manuscript(cancers-1186122). We hope that these changes make our manuscript more compliant with the formatting and content guidelines of Cancers.
With these changes to our final manuscript, we hereby resubmit it for an evaluation. Thank you once again for your consideration of our paper.
Reviewer #1
Comment-1: In figure 4, what does the protein expression of these two integrins look like in the normal liver? Is it restricted to certain cells (cholangiocytes) or more generally?
Response:
I really appreciate your valuable comments. The following sentence was added based on the reviewer's suggestions in Results.
Immunostaining confirmed that ITGB1 was expressed in normal cholangiocytes and hepatocytes (Fig. S1). Compared with ITGB1, expression of ITGA6 was hardly observed (Fig. S1).
Comment-2: Throughout the study (and particularly in figure 8) its not clear whether B1A6 are co-expressed in the same cells. Can the authors show co-localization in cells or Co-IP...the assumption is in this paper that these receptors are a heterodimer, but that is not proven in this context.
Response: We recognize that the reviewer’s comment is important issue in this study. It is a well-known that integrin function by forming heterodimers in cells, integrin B1 forms a heterodimer with A6. We have no means of analyzing whether each molecule forms a heterodimer (ITGA6 and ITGB1) in the cell. As a result of immunostaining, it has been confirmed that both proteins (ITGA6 and ITGB1) are expressed in cancer cells.
From your valuable comment, I examined the expression levels of other integrin-family genes in ICC specimens by in silico analysis. I obtained some interesting knowledge, so I'd like to show it in a revised version.
We also analyzed expression levels of other ITGs that form heterodimers with ITGB1 in ICC clinical specimens using GEPIA2 database (Fig.S9). The expression of some ITGs e.g., ITGAV, ITGA2, ITGA3, ITGA5, and ITGA7 were upregulated in ICC tissues. Also, upregulation of ITGB4 (form heterodimers with ITGA6) was detected in ICC tissues. It was suggested that not only the ITGA6/ITGB1 dimer but also the ITGs that form dimers with each of these may play an important role in ICC.
Comment-3: What's not clear to me is that in figure 9 whether integrins can affect XTT, how do you know that this isn't mitochondrial dysfunction - are there XTT.
Response: I'm sorry, I don't fully understand the meaning of your comments. Integrins (ITGA6 and ITGB1) knockout did not show any significant inhibition of cell proliferation abilities. However, statistical analysis showed significant differences in cell proliferation inhibitory effects.
Comment-4: In 10A, the Y axis is labelled with A6, presumably this should be B1?
Response: Thanks for pointing out. I apologize for the mistake. I changed the text.
Comment-5: The proliferation and migration effects are different between IntergrinA6 and B1, have you evaluated whether the downstream changes are the same with KD of the individual changes?
Response: Although it is important for reviewer's point out, comprehensive gene expression analysis using cells in which genes have been knocked down will be an issue for the future.
Comment-6: What happens if both of the integrins are knocked down at the same time, are the phenotypes additive?
Response: According to the reviewer’s suggestion, we reanalyzed using cells in which two genes were knocked down at the same time. Knocking down two molecules at the same time enhanced the tumor suppressor effects. This data is shown in the Figure S4, and added the following sentences to the Results.
Furthermore, both integrins (ITGA6 and ITGB1) were knocked down at the same time to examine the tumor suppressor effects, e.g., cell proliferation and migration, and invasive abilities. By knocking down both integrins, the malignant phenotypes of cancer cells were further controlled (Figure S4).
Comment-7: Can you conform whether SP1 protein is expressed in tumour tissue?
Response: I deeply appreciate your advice. As the reviewer pointed out, in addition to Figure 12, I added Figure S7, which includes HE staining, to make it easier to understand the expression of SP1 protein in the cancer tissue.
Sincerely yours,
Naohiko Seki, Ph.D.
Department of Functional Genomics
Chiba University Graduate School of Medicine
1-8-1 Inohana, Chuo-ku,
Chiba 260-8670, Japan
Phone: +81-43-226-2971
Fax: +81-43-227-3442
e-mail: naoseki@faculty.chiba-u.jp

Reviewer 2 Report
Hozaka et al. provide an interesting study of expression of miR-29-3p-family in association with ITGA6/ITGB1 in intrahepatic cholangiocarcinoma.
This study has been performed on two human cholangiocarcnoma cell cultures. The experiments are well designed and the results are clear. Although, the experimental design is not innovative. The effect of the miR-29-family and Integrin gene has been studied in hepatic stellate cell activation (Sekiya Y, et al. Biochem Biophys Res Commun. 2011; 412 (1): 74-79). The α6β4 (β4) integrin is a biliary-type integrin that is expressed on normal and proliferating bile ducts and cholangiocarcinomas, but not on hepatocytes and hepatocellular carcinoma (HCC) Volpes R., van den Oord J.J., Desmet V.J. Integrins as differential cell lineage markers of primary liver tumors. Am. J. Pathol. 1993;142:1483–1492.
However, to the best of my knowledge, these results have not been previously described in human intrahepatic cholangiocarcinoma cell lines.
Although there are no clinical data in this work, these results could represents a good clue in the characterization of the different type of cholangiocarcinoma. In particular the investigation of miR-29-3p-family regulation in cholangiocarcinoma associated to previous liver pathology (chronic active hepatitis and/or cirrhosis) or in cholangiocarcinoma developed in normal liver could represent a good tool to understand both the pathogenesis and the different aggressivenes of these group of cancers. Furthermore the possibility to define new therapeutic strategies is an important objective in these cancers.
Minors:
There is an incorrect reference, line 84, reference [23].
Author Response
Revise letter(Manuscript ID: cancers-1186122)
May 28, 2021
Dr. Takahiro Kodama
Guest Editor
Department of Gastroenterology and Hepatology,
Osaka University Graduate School of Medicine,
Osaka, Japan
Dear Dr. Kodama:
Thank you for your constructive comments and suggestions. We believe that our manuscript has been greatly improved and is now suitable for publication in Cancers. We agree with your and the reviewers’ comments and suggestions. All changes are highlighted in the revised manuscript(cancers-1186122). We hope that these changes make our manuscript more compliant with the formatting and content guidelines of Cancers.
With these changes to our final manuscript, we hereby resubmit it for an evaluation. Thank you once again for your consideration of our paper.
Reviewer #2
This study has been performed on two human cholangiocarcinoma cell cultures. The experiments are well designed and the results are clear. Although, the experimental design is not innovative. The effect of the miR-29-family and Integrin gene has been studied in hepatic stellate cell activation (Sekiya Y, et al. Biochem Biophys Res Commun. 2011; 412 (1): 74-79). The α6β4 (β4) integrin is a biliary-type integrin that is expressed on normal and proliferating bile ducts and cholangiocarcinomas, but not on hepatocytes and hepatocellular carcinoma (HCC) Volpes R., van den Oord J.J., Desmet V.J. Integrins as differential cell lineage markers of primary liver tumors. Am. J. Pathol. 1993;142:1483–1492.
However, to the best of my knowledge, these results have not been previously described in human intrahepatic cholangiocarcinoma cell lines.
Although there are no clinical data in this work, these results could represents a good clue in the characterization of the different type of cholangiocarcinoma. In particular the investigation of miR-29-3p-family regulation in cholangiocarcinoma associated to previous liver pathology (chronic active hepatitis and/or cirrhosis) or in cholangiocarcinoma developed in normal liver could represent a good tool to understand both the pathogenesis and the different aggressiveness of these group of cancers. Furthermore the possibility to define new therapeutic strategies is an important objective in these cancers.
Response: I sincerely would like to thank meaningful comments of reviewer. I referred previous study and added the following sentences in Discussion.
Additionally, liver fibrosis is featured by an abnormal accumulation of extracellular matrix (ECM) and is a common feature of chronic liver damage. The hepatic stellate cell (HSC) is involved in liver fibrosis, which is the formation of scar tissue in response to liver damage. Previous study showed that miR-29b was downregulated during HSC activation [40]. Ectopic expression of miR-29b inhibited several genes involved in HSC activation, e.g., COL1A1, COL1A2, DDR2, FN1, and ITGB1 [41].
Minors:
There is an incorrect reference, line 84, reference [23].
Response: Thank you for pointing out. I fixed.
Sincerely yours,
Naohiko Seki, Ph.D.
Department of Functional Genomics
Chiba University Graduate School of Medicine
1-8-1 Inohana, Chuo-ku,
Chiba 260-8670, Japan
Phone: +81-43-226-2971
Fax: +81-43-227-3442
e-mail: naoseki@faculty.chiba-u.jp

Reviewer 3 Report
The paper is of interest.
Some minor hints:
- I would go for a better introduction on molecular features of CCA, dividing molecular alterations for different primary sites. It may be also a graph and an example is here: Cancer Management Research. 2018 Dec 28;11:379-388.doi: 10.2147/CMAR.S157156. eCollection 2019. Perhaps a graphical representation of pathways involved may be useful
- Please discuss any potential existing clinical trials (phase I perhaps) and discuss any clinical future development (how would you design a phase II study for example? Any ideas on future development?
Author Response
Revise letter(Manuscript ID: cancers-1186122)
May 28, 2021
Dr. Takahiro Kodama
Guest Editor
Department of Gastroenterology and Hepatology,
Osaka University Graduate School of Medicine,
Osaka, Japan
Dear Dr. Kodama:
Thank you for your constructive comments and suggestions. We believe that our manuscript has been greatly improved and is now suitable for publication in Cancers. We agree with your and the reviewers’ comments and suggestions. All changes are highlighted in the revised manuscript(cancers-1186122). We hope that these changes make our manuscript more compliant with the formatting and content guidelines of Cancers.
With these changes to our final manuscript, we hereby resubmit it for an evaluation. Thank you once again for your consideration of our paper.
Reviewer #3
Some minor hints:
- I would go for a better introduction on molecular features of CCA, dividing molecular alterations for different primary sites. It may be also a graph and an example is here: Cancer Management Research. 2018 Dec 28;11:379-388.doi: 10.2147/CMAR.S157156. eCollection 2019. Perhaps a graphical representation of pathways involved may be useful.
- Please discuss any potential existing clinical trials (phase I perhaps) and discuss any clinical future development (how would you design a phase II study for example? Any ideas on future development?
Response: I sincerely would like to thank meaningful comments of reviewer. I referred previous study and added the following sentences in Discussion.
Cholangiocarcinoma is anatomically divided into ICC and extrahepatic cholangiocarcinoma. From cancer genomic analysis, each cancer recognizes characteristic gene mutations, suggesting molecular biological differences [1-3].
At this time, no molecular-targeted drugs targeting integrin-family are used for ICC. Searching for specific integrin-mediated molecular pathways in ICC will help develop new therapeutic agents.
Sincerely yours,
Naohiko Seki, Ph.D.
Department of Functional Genomics
Chiba University Graduate School of Medicine
1-8-1 Inohana, Chuo-ku,
Chiba 260-8670, Japan
Phone: +81-43-226-2971
Fax: +81-43-227-3442
e-mail: naoseki@faculty.chiba-u.jp

Round 2
Reviewer 1 Report
No further comments